# The Novel Porcine Parvoviruses: Current State of Knowledge and Their Possible Implications in Clinical Syndromes in Pigs

**DOI:** 10.3390/v15122398

**Published:** 2023-12-09

**Authors:** Diana S. Vargas-Bermudez, Jose Dario Mogollon, Camila Franco-Rodriguez, Jairo Jaime

**Affiliations:** Universidad Nacional de Colombia, Sede Bogotá, Facultad de Medicina Veterinaria y de Zootecnia, Departamento de Salud Animal, Centro de Investigación en Infectología e Inmunología Veterinaria (CI3V), Carrera 30 No. 45-03, Bogotá 111321, CP, Colombia; dsvargasb@unal.edu.co (D.S.V.-B.); josedmogollon@yahoo.es (J.D.M.); macfrancoro@unal.edu.co (C.F.-R.)

**Keywords:** parvoviruses, novel porcine parvoviruses (nPPVs), porcine parvoviruses (PPVs), coinfections, porcine respiratory disease complex (PRDC), porcine reproductive failure (PRF)

## Abstract

Parvoviruses (PVs) affect various animal species causing different diseases. To date, eight different porcine parvoviruses (PPV1 through PPV8) are recognized in the swine population, all of which are distributed among subfamilies and genera of the *Parvoviridae* family. PPV1 is the oldest and is recognized as the primary agent of SMEDI, while the rest of the PPVs (PPV2 through PPV8) are called novel PPVs (nPPVs). The pathogenesis of nPPVs is still undefined, and whether these viruses are putative disease agents is unknown. Structurally, the PPVs are very similar; the differences occur mainly at the level of their genomes (ssDNA), where there is variation in the number and location of the coding genes. Additionally, it is considered that the genome of PVs has mutation rates similar to those of ssRNA viruses, that is, in the order of 10^−5^–10^−4^ nucleotide/substitution/year. These mutations manifest mainly in the VP protein, constituting the viral capsid, affecting virulence, tropism, and viral antigenicity. For nPPVs, mutation rates have already been established that are similar to those already described; however, within this group of viruses, the highest mutation rate has been reported for PPV7. In addition to the mutations, recombinations are also reported, mainly in PPV2, PPV3, and PPV7; these have been found between strains of domestic pigs and wild boars and in a more significant proportion in VP sequences. Regarding affinity for cell types, nPPVs have been detected with variable prevalence in different types of organs and tissues; this has led to the suggestion that they have a broad tropism, although proportionally more have been found in lung and lymphoid tissue such as spleen, tonsils, and lymph nodes. Regarding their epidemiology, nPPVs are present on all continents (except PPV8, only in Asia), and within pig farms, the highest prevalences detecting viral genomes have been seen in the fattener and finishing groups. The relationship between nPPVs and clinical manifestations has been complicated to establish. However, there is already some evidence that establishes associations. One of them is PPV2 with porcine respiratory disease complex (PRDC), where causality tests (PCR, ISH, and histopathology) lead to proposing the PPV2 virus as a possible agent involved in this syndrome. With the other nPPVs, there is still no clear association with any pathology. These have been detected in different systems (respiratory, reproductive, gastrointestinal, urinary, and nervous), and there is still insufficient evidence to classify them as disease-causing agents. In this regard, nPPVs (except PPV8) have been found to cause porcine reproductive failure (PRF), with the most prevalent being PPV4, PPV6, and PPV7. In the case of PRDC, nPPVs have also been detected, with PPV2 having the highest viral loads in the lungs of affected pigs. Regarding coinfections, nPPVs have been detected in concurrence in healthy and sick pigs, with primary PRDC and PRF viruses such as PCV2, PCV3, and PRRSV. The effect of these coinfections is not apparent; it is unknown whether they favor the replication of the primary agents, the severity of the clinical manifestations, or have no effect. The most significant limitation in the study of nPPVs is that their isolation has been impossible; therefore, there are no studies on their pathogenesis both in vitro and in vivo. For all of the above, it is necessary to propose basic and applied research on nPPVs to establish if they are putative disease agents, establish their effect on coinfections, and measure their impact on swine production.

## 1. Introduction

Parvoviruses (PVs) constitute a large family of viruses that infect different animals, both vertebrates and invertebrates [1,2]. In recent years, new PVs have been described due to advances in sequencing techniques (high-throughput) and metagenomic analyses [3]. In mammals, PVs are related to multiple clinical conditions such as reproductive diseases in pigs [4], enteritis in dogs [5], panleukopenia in cats [6], nervous disease and immune complex-mediated vasculitis in humans [7], hepatitis in horses [8], and respiratory disease also in humans [9]. In the swine population, eight porcine parvoviruses (PPVs) have been described (PPV1 through PPV8) that are distributed within different subfamilies and viral genera [10,11,12,13,14,15], particularly PPV2 through PPV8 are called novel PPVs (nPPVs). Although the complete genome sequences for all PPVs are reported, it is still unknown whether nPPVs cause or promote the presentation of diseases [3]. PPV1 has been circulating since the 1960s, and it is associated with porcine reproductive failure (PRF) as a primary agent of SMEDI, which causes stillbirth, mummification, embryonic death, and infertility [16]. PPV2 was reported in 2001 and proposed as a possible primary pathogen of the porcine respiratory disease complex (PRDC) [17,18]. PPV3 through PPV8 do not have a defined pathogenesis [3] and their participation in causing disease is debated. This discussion arises because nPPVs have been detected in different types of samples and tissues from healthy pigs and pigs with clinical diseases (primarily respiratory) [13], which has led to the proposal that they are commensal [12,19] or are part of the normal virome of pigs and do not generate disease [3]. The fundamental limitation to establishing whether nPPVs are pathogenic is that they have not been isolated yet [10,12], making pathogenesis assays in vitro and in vivo impossible. Furthermore, it was proposed that nPPVs could establish persistent infections, as has already been determined in other members of the *Parvoviridae* family [20]. 

Although it was found that PPVs share a joint genetic base, that is, they have a ssDNA genome that has two open reading frames (ORFs), there are differences among PPVs, such as the presence of additional ORFs or the use of different transcription strategies [21]. These differences cause their biological activities to vary. At the field level, nPPVs have been detected on swine farms worldwide, at all stages of the pig production cycle, and in a wide variety of tissues and samples [22,23,24]. Regarding the origin of nPPVs, it was established that there is genetic divergence among them due to recombination events, positive selection, and adaptation to new hosts (transmission from wild to domestic pigs) [25]. This review aims to compile the existing information on nPPVs and propose their possible role as putative agents or participants in clinical manifestations, that is, in diseases.

## 2. Classification and Molecular Organization of Porcine Parvovirus

The *Parvoviridae* family harbors small viruses of approximately 25 nm in diameter that are structurally non-enveloped and have a linear ssDNA genome with a length ranging between 4 and 6 Kb [21]. This family is divided into three subfamilies [1]: (i) *Densovirinae*, which infects invertebrates and which in turn is divided into 11 genera *(Diciambidensovirus*, *Muscodensovirus*, *Tetuambidensovirus*, *Scindoambidensovirus*, *Blattambidensovirus*, *Hemiambidensovirus*, *Aquambidensovirus*, *Protoambidensovirus, Miniambidensovirus*, *Iteradensovirus*, and *Pefuambidensovirus)*; (ii) *Parvovirinae,* which infects vertebrates and is divided into 11 genera (*Protoparvovirus*, *Amdoparvovirus*, *Bocaparvovirus, Aveparvovirus*, *Tetraparvovirus*, *Dependoparvovirus*, *Copiparvovirus, Artiparvovirus*, *Loriparvovirus*, *Erythroparvovirus*, *and Zanderparvovirus*); and (iii) the more recent subfamily *Hamaparvovirinae* divided into 5 genera (*Pennstylhamaparvovirus, Brevihamaparvovirus, Hepanhamaparvovirus, Chaphamaparvovirus* and *Ichthamaparvovirus*) that infect both vertebrates and invertebrates [1,26]. In the swine population, eight genetically divergent species of PPVs (PPV1 through PPV8) have been detected [10,11]. Of these eight species, seven belong to the subfamily *Parvovirinae*. Among genera, they are distributed as follows: PPV1 and PPV8 belong to the genus *Protoparvovirus*, PPV2 and PPV3 to *Tetraparvovirus*, PPV4 and PPV6 to *Copiparvovirus*, and PPV5 remains unclassified by the ICTV [1]. On the other hand, PPV7 belongs to the new *Hamaparvoviriane* subfamily and the *Chaphamaparvovirus* genus [27]. Figure 1 shows the classification of the PPVs according to their phylogeny, and Table 1 describes the characteristics of the PPVs based on the length of the genome, the species, and the genus to which they belong.

The PV-ssDNA has particularities, such as forming palindromic sequences (hairpins) of 120–200 bp at its two ends. Furthermore, it contains two or three ORFs. The ORF1 or left-ORF is located close to the 5’ end and encodes non-structural proteins (NS1 and NS2) or regulatory proteins participating in the viral replication process. NS1 has helicase and nickase activity and can induce apoptosis and cell lysis [29]. The NS2 protein is translated after alternative splicing of the transcribed mRNA, and its function is related to participation in the process of viral egress from the nucleus through its interaction with the nuclear export factor CRM1 [30]. In the case of PPV1, NS2 has been reported to suppress type I interferon (INF-I) activity [31]. Another property of the ORF1 product (NS protein) is that it is highly conserved among the PV subfamilies, using its amino acid (aa) sequence as a criterion for classifying PVs into genera [21]. The ICTV established that to classify PVs, >85% identity in the aa sequence of the NS protein is necessary, allowing within the same genus [1,2]. The ORF2, or right-ORF, is located close to the 3′ end, encodes up to four structural proteins (VPs), a quantity that varies depending on the viral genera [21], and their primary function is to be part of the viral capsid. VPs have two highly conserved regions among different viral genera: the phospholipase A_2_ (APLA_2_) domain, necessary for viral infectivity [32], and a domain that binds calcium (Ca_2_-binding loop) with nuclear localization signals. Some particularities have been reported comparing these capsid domains between the different PPVs: the Ca2-binding loop has the YXGXG sequence in PPV1 to PPV3 [10] and PPV5 [11]; YXGXR for PPV6 [12]; and YXGXXG for PPV7. The APLA_2_ motif maintains the HDXXY sequence in PPV1 to PPV3, PPV5111321, and PPV6 [11,12]. An exception to the above is PPV7, which does not possess APLA_2_ in VP (pattern for the *Hamaparvovirinae* subfamily) [21], and PPV4 does not possess either of the two mentioned domains (Ca_2_-binding loop and APLA_2_). Another peculiarity of PPV4 is that its genome has a circular (cssDNA) or head-to-tail concatemeric shape, unlike the rest of PPVs with a linear head-to-head or tail-to-tail structure. This PPV4-cssDNA has suggested that it could cause persistence in infected cells [33]. In addition to the differences in the VP domains and the shape of the genome, in some members of the *Parvoviridae* family, small proteins or accessory proteins have been detected as a result of overlapping ORFs in the expression cassettes or positioned between ORF1 and ORF2 and that have been named ORF3 [26,34]. In this sense, PPV1 has a short ORF that overlaps ORF 2 and encodes the small alternatively translated protein (SAT), whose presumed function is to induce stress in the endoplasmic reticulum, facilitating cell lysis [34]. In PPV4, an ORF3 located between ORF1 and ORF2 has been reported and encodes a 204 aa protein, whose function is still unknown [35]. At the same time, in PPV7, a predicted minor ORF overlaps the rep gene and encodes a 220 aa protein, which has a regulatory function [27].

**Table 1 viruses-15-02398-t001:** Main characteristics of porcine parvoviruses (PPV1 through PPV8) based on the ITCV classification, genome length, ORFs, year of discovery, and the reported evolution rate.

Virus (GenBank Accession Number)	ICTV Current Classification (2022)	Genome Size (nt ^¶^)	ORF 1 (aa ^#^)	ORF 2 (aa)	Year of the Discovery	Evolutionary Rates (Substitutions/Site/Year) *	Reference
PPV1 (L23427)	Protoparvovirus Ungulate 1	5075	662	729	1965	6.22 × 10^−5^	[36]
PPV2 (KM926355)	Tetraparvovirus Ungulate 3	5444	662	1032	2001	1.35 × 10^−4^	[37]
PPV3 (EU200677)	Tetraparvovirus Ungulate 2	5114	637	926	2008	8.16 × 10^−4^	[10]
PPV4 (FJ872544)	Copiparvovirus Ungulate 2	5905	588	728	2010	4.70 × 10^−4^	[33]
PPV5 (JX896318)	Not yet classified	5805	601	991	2013	6.95 × 10^−5^	[38]
PPV6 (KY094494)	Copiparvovirus Ungulate 4	6148	662	1189	2014	4.90 × 10^−4^	[12]
PPV7 (MG902949)	Chaphamaparvovirus Ungulate 1	4103	672	469	2015	4.85 × 10^−3^	[14]
PPV8 (OP021638)	Not yet classified	4380	601	701	2022	Not reported	[15]

^¶^ nt: nucleotides; ^#^ aa: amino acids; * Evolutionary rates taken from [39].

Concerning the transcription and translation strategies employed by PVs, it was described differences depending on the viral genus. Some PVs use a single promoter to express the coding regions for NS and VP proteins, while others use two, one for each protein [21]. Likewise, many mRNA-PVs perform alternative splicing or use alternative star codons for VP expression [21]. The above leads to PVs that use these alternative mechanisms, expressing isoforms of proteins with different functions [40]. Concretizing the previous concepts for PPVs, PPV1 has two promoters: P4, which regulates the expression of three NS proteins (NS1, NS2, and NS3), and P40, which controls the expression of VP proteins (VP1, VP2, and VP3) [41]. Regarding the latter, in PPV1, VP1 and VP2 are translated from a nested set of coding sequences; mainly, VP2 is generated by splicing the mRNA that codes for VP1. In contrast, VP3 is produced by a post-translational modification of VP2 [42]. Regarding nPPVs, some particularities were established. Through an artificial replication system using PARV4 (human parvovirus 4), it was possible to understand the transcription strategies of the *Tetraparvovirus* (PPV2 and PPV3). Thus, this genus uses two promoters (P6 and P38) that transcribe mRNAs for NS and VP, respectively [40]. In PPV3, it was also detected that ORF2, besides encoding VP proteins, also codes for a small conserved putative protein [10]. At least three potential promoters were determined in silico analysis for PPV7 (*Chaphamaparvovirus*). However, it is still unknown how these regulate the transcription of viral genes [27]. The particular transcription and protein expression strategies are still novel for the rest of nPPVs (PPV4 to PPV6 and PPV8). Regarding the capsid of the PVs, it has an icosahedral symmetry of ~260 Å in diameter. 

The capsid consists of 60 individual capsomers (VP proteins) [21], where each contains eight antiparallel β-strands, along with one α-helix and four loops [43]. The loops exposed on the surface contain the aa sequences with the most significant variability; these correspond to the vertices of the loops and have been named variable regions (VR) that differ between members of the same genus and different PVs [21]. In *Protoparvoviruses*, 9–10 VRs have been identified, but for the other genera, they are still unknown [21]. It is essential to point out that, despite the high variability in the aa sequence of the VPs, the morphology of the capsid between them is conserved [21,27].

## 3. Evolution of Porcine Parvoviruses 

For PVs, evolutionary rates have been reported similar to those detected for RNA viruses [44]. This high degree of variability has been related to (i) the rolling hairpin replication system of PV-ssDNA [45], (ii) the use of replicative enzymes of parvoviral origin with lower fidelity than cellular ones [25,46], and (iii) de novo parvoviral mutated genomes cannot be repaired by host exonucleases because they do not possess methylation patterns [47].

### 3.1. Mutations in Porcine Parvovirus

Mutation rates in PVs have been detected in higher proportions in genes that encode capsid proteins (VP) versus genes that encode NS [48,49]. In PPV1, in both domestic and wild pigs, mutation rates of 3 − 5 × 10^−4^ nucleotide/substitutions/year (nsy) in the VPs-encoding genes were reported, while for NS, they were 10^−5^ nsy [48,49]. This high variability in the PPV1-VP sequence is what has served as the basis for its phylogenetic classification with several proposals: (i) into two genetic lineages (G1 and G2) [50]; (ii) between 6 and 7 clusters (A to H) [48]; (iii) in four lineages (1 to 4) [51]; and more recently (iv) in four genotypes (PPV1a to d) [52]. The PPV1-VP mutations found are mainly in the loops on the capsid’s surface; these loops bind to the host cell receptor and act as antigenic targets [49]. Additionally, this heterogeneity in PPV1-VP has been associated with effects on virulence [53], tropism [41], and antigenicity [54]. In the last two decades, divergences between PPV1 strains have increased, leading to the proposal that this is the product of an adaptation of PPV1 to the vaccines used, generating escape mutants in populations with partial immunity (non-sterilizing) against the virus [53]. The above does not ignore that vaccination against PPV1 has served to limit its transmission and control viral variability, which is explained by the fact that the variability is more significant in wild pigs (not vaccinated) [48]. Regarding nPPVs and their evolutionary rate, it is essential to note that this is not constant. For PPV2 to PPV6-VP, between 10^−5^ and 10^−4^ nsy has been reported (Table 1) [39,55], while for PPV7, the highest rate among all PPVs has been detected with 2.19 × 10^−3^ nsy for VP and 8.01 × 10^−4^ nsy for NS [39,56]. Regarding the phylogenetic analyses of nPVVs, it should be emphasized that they are limited due to the low number of sequences reported to date, mainly due to their low detection. Therefore, as more sequences become available from diverse geographical origins, better classifications of these viruses can be made and proposed. For PPV2, initial phylogenetic studies, based on partial VP-nt sequences, proposed a classification into seven clades (A to G) [57]. Subsequently, with analysis of complete VP-nt sequences, classification was proposed into two clades: PPV2-Clade A, harboring strains from Myanmar in 2001, some from China and Hungary; and PPV2-Clade B harboring strains from China and the rest of the world [25,58]. Analyzes based on aa have proposed that some residues in the PPV2-VP region may be critical for differentiation between clades [58]. In the case of PPV3, European studies have proposed a classification into four clades (A to D), each exhibiting a specific VP-aa pattern [25]. This classification determined that strains reported in China are located in the four clades, while those from North and South America are in clade C, and those from Europe are in C and D [59]. Another classification proposed for PPV3 [24] divides it into three clades (1 to 3), where 1 and 2 contain sequences only from China and 3 from other regions of the world. The classification of PPV4 has been proposed in two clades (A and B or also 1 and 2) [58], where A or 1 corresponds to strains from Europe reported between 2004 and 2011, while clade B or 2 groups strains from other parts of the world [24]. The PPV5 has not yet been classified into clades or clusters [60], while for PPV6, two classifications have been proposed, one in four clusters [61] and another in three clades (1 to 3) [60]. For PPV7, the first classifications were supported on complete genome sequences, and two clades (1 and 2) were established [56]. However, it is important to note, as already mentioned, that the PPV7-VP region presents the most significant variability between the PPVs, explained by insertions (15nt, 5aa), finding that the length of VP can vary between 1405 and 1425 aa [62]. In this way, the phylogenetic analyses of PPV7-VP propose two classifications, the first corresponding to two groups (group 1 with and group 2 without insertion) [62] and the second in three clades (1 to 3) [60]. Recently, with the availability of a more significant number of PPV7-full genome sequences, a classification into six clades (a to f) has been proposed [63]. Independent of the classification, it is essential to note that this variability in PPV7-VP could affect adsorption, antigenicity, and immune responses [56,62], similar to what has been described for PPV1.

### 3.2. Recombination of Porcine Parvoviruses

Recombination events occur in the genomes of PVs at the interspecies level [64]; these are associated with replicative machinery that would facilitate this event by copy–choice mechanisms [46]. A viral recombination event requires circumstances that are not frequent, such as the concurrent infection of a host and a cell with viruses that can carry out this exchange [64]. For example, in humans, factors that favor the recombination of human parvoviruses (HPVs) have been determined, such as the persistence of the viral ssDNA and the concurrent presence of several coinfecting genotypes [65]. In the case of PPVs, the factors that support their recombination are unknown. However, some have been proposed mainly related to the characteristics of swine production systems such as high population densities that have close contact between pigs, the mixture of animals from different origins (farms) for growing, partial or incomplete herd immunity, contact between domestic and wild pigs, among others [25,64,66]. For PPV1, recombination events have been described between low- and high-virulence strains [67]. The possible explanation for this is when a pig does not generate sterilizing immunity (mainly of vaccine origin), and there are persistent viruses that can recombine with new others [64]. For PPV7, chimeric strains are coming from the recombination between strains that circulate in wild pigs and those in domestic pigs [22,66]. Particularly in Republic of Korea, the PPV7-KF4 strain was identified as a recombinant with the 17KWB09 strain of the Korean wild boar [66]. In China in 2021, the PPV7-HBTZ20180519-152 strain was reported, a recombinant generated between the main parental virus JX15 and strain minor parental JX38, both isolated from Chinese wild boars in 2015; this led to the proposal that natural recombination of PPV7 occurred in wild boars and could be beneficial for the transmission of PPV from wild boars to domestic pigs [22]. In PPV2 and PPV3, recombination patterns were established at the VP gene coding and between domestic and wild pigs [25,55]. In Romania, recombination signals for PPV2 were detected along all regions of the VP gene, while in the case of PPV3, only in the middle of the gene. The parental strains of the PPV2 and PPV3 recombinants seem mainly to be of mixed origin (domestic and wild pigs) and, in a few cases, only with strains detected in one of the species. Recombination events involving the same parental strains were detected in wild boar PPV2 strains [25]. In a study detecting PPV2 to PPV4 in sample tissues and serum from Croatia, Poland, Serbia, Hungary, and Romania, collected between 2006 and 2011 from domestic pigs, strong recombination signals were detected in the PPV2-VP dataset using GARD and Splits Tree network analyses. These events were detected both within and between clades, within and between countries, and also between domestic and wild (wild boar) strains [55]. For PPV4, PPV5, and PPV6, to date, no recombination events have been reported [22]. 

## 4. Epidemiology of Porcine Parvoviruses

### 4.1. Distribution of Porcine Parvoviruses

PPV1 was first reported in Germany in 1965 as a contaminant of porcine cell cultures [68]. This virus is distributed worldwide and associated with PRF as one of the primary agents of SMEDI [69]. PPV1 has been controlled by vaccinating replacement gilts and sows [16]. PPV2 was discovered incidentally in 2001 in Myanmar from pig sera collected to detect the hepatitis E virus [70]. Ten years later, in China, it was found again in farms with a history of porcine respiratory and reproductive syndrome virus (PRRSV) and post-weaning multisystemic wasting disease (PMWS) in pigs that had “High fever disease” [71]. Subsequently, PPV2 was detected in Asia [57,71,72], Europe [24,55,73,74], Africa [75], Oceania [76], North America [77], Central America [78], and South America [37]. Likewise, PPV2 was also reported in both domestic and wild pigs [25]. PPV3, also known as porcine PARV4, hokovirus, or partetravirus, was reported for the first time in 2008 in Hong Kong from samples of different types of tissues collected in slaughter plants with a prevalence of 44.4% [10]. Subsequently, it was detected in Europe [25,73,79,80,81], Asia [82], North America [83,84], and South America [85]. It is essential to highlight that PPV3 was found in high prevalence (20–54%) in wild pigs from some European countries [25,79,81]; likewise, in both young and adult wild pigs [86]. As for PPV4, it was detected for the first time in the USA in 2010 from the lungs of pigs coinfected with porcine circovirus type 2 (PCV2) [33]. Later, it was reported in other countries, such as Sweden, from lymph nodes in a farm with PMWS [87], and in China, both in healthy pigs and in pigs with symptoms (tremors, fever, abortions, death, and testicular atrophy); additionally, these pigs were determined to have PPV4/PCV2 and PPV4/TTV1-2 (porcine torquetenovirus 1 and 2) coinfections [88]. Currently, PPV4 is considered to be present in different countries in Europe [59,73,80], Asia [60], Africa [75,89], and North and South America [78,90]. It has also been found in wild pigs in Europe and Africa [25,89]. PPV5 was detected for the first time in 2013 in the USA from lung samples [38] and then in feces from pigs with different syndromes such as PRDC, gastro-intestinal, and PRF, found mainly in grow-finish pigs, and finishing pigs and in coinfection with PPV5/PPV4 [11]. Subsequently, PPV5 was detected in the lungs of pigs with porcine circovirus-associated disease (PCVAD) [13] and PMWS [91]. The distribution of PPV5 is more restricted and has been detected in China [91], Republic of Korea [60], Poland [80], Mexico [78], Brazil [92], and Colombia (own unpublished data). PPV6 was reported in 2014 in China from aborted fetuses, piglets, and breeding sows on farms with PRF and healthy ones [12]. It was subsequently detected in the USA and Mexico from sera [93], in Spain in healthy pigs [94], in Poland and Republic of Korea from pigs of different ages [60,61,95], in Brazil in pork chops [92], and in Colombia from sera of replacement gilts (own unpublished data). PPV7 was identified in 2016 in the USA through metagenomic analysis of stool samples [14]. Subsequently, it was detected in different continents [23,62,76,96] from different types of samples with varied prevalences and in wild pigs [22,66]. Finally, PPV8 was reported in China in a retrospective study (1998–2021) carried out on lung samples from pigs with symptoms of fever or respiratory signs, finding a prevalence of 17.4% [15]. The distribution of nPPVs at a global level is discriminated in Figure 2. Phyloevolutionary studies have determined that these nPPVs emerged before the date of their first reports and have been determined as follows: PPV1 since 1855 [52], PPV2 since the 1920s, PPV3 since the 1930s, PPV4 since 1980 [55], and PPV7 since 2004 [56].

### 4.2. Prevalence by Age Groups of Novel Porcine Parvovirus

Prevalence studies of nPPVs (PPV2 through PPV7) by age groups in production pigs have shown that this is higher in fatteners and finishing pigs [11,22,23,24,95] and lower in suckling and weaned pigs [24,77,80,83,95]. This low prevalence in pigs under nine weeks of age suggests probable adequate protection through passive immunity [17]. Particularly for PPV2, high titers of passively acquired antibodies (Abs) have been detected between 2 and 6 weeks of age, and subsequently, an increase in viral loads between weeks 9 and 13 was associated with some non-specific symptoms [80]. To date, the dynamics of maternal Abs to PPV3 to PPV8 are unknown, but it is suggested that it may have a similar profile to PPV2 [22]. It is also important to note that among the nPPVs, PPV6 is the one most detected in piglets [95].

## 5. Tropism of Porcine Parvoviruses 

The fact that PPVs have been detected in different types of samples (tissues) and with varied prevalences suggests that they have a broad tropism. For PVV1, tropism both in organs (in vivo) and at the level of cell cultures (in vitro) has been widely studied [53]. PPV1 has a tropism for macrophages that migrate to the placenta, where the virus replicates and infects the fetus, disseminating within it and affecting various organs, mainly the lung tissue [4]. Regarding the tropism of nPPVs, these viruses have been detected in high loads in the serum [13,73,95] and the tissue, mainly in lung and lymphoid organs such as the spleen, tonsils, and lymph nodes [10,24,73]. At the respiratory system, the majority of nPPVs have been detected in nasal swabs (PPV2 to PPV 4 and PPV7) [10,24,97], lung tissue (PPV2 to PPV8) [13], bronchoalveolar lavages [14,33], and in bronchial lymph nodes [24]. In pigs with PRDC, the most detected nPPV is PPV2, with prevalences reaching 75% [24] and, to a lesser extent, PPV3 to PPV6 [13,24]. PPV2 has also been detected in high prevalence in the tonsils [24,57,74] and heart. In the gastrointestinal system, PPV2 was detected in samples of both oral fluid and feces in healthy pigs [95], while evaluation of pools of large intestine and feces revealed the presence of PPV2 to PPV4 and PPV6 [98] (Table 2). PPV3 and PPV7 were detected in high prevalence in the liver, the former in both domestic and wild pigs [10,79] while the latter only in domestic pigs [99]. In the urinary system, PPV2 to PPV4 has been found in the kidney [24]. In the reproductive system, it should be noted that in sows with PRF, all PPVs have been detected (except PPV8), particularly in the hearts of aborted fetuses and in those coinfected or not with PCV2 [24,78] (Table 2 and Table 3). Breaking down the above, in aborted fetuses, PPV2 has been detected with varied prevalences. For example, in a low proportion (1.3%) it has been detected in thoracic fluids [73,77] and a very high proportion (93%) in hearts [78]. For PPV3 and PPV5, the results are contradictory since there are studies where they were not detected in fetal samples [10,73], while in others, their prevalence reached 30% [78]. For PPV4, PPV6, and PPV7, studies report their high prevalence in aborted fetuses [12,73,100], while PPV4 and PPV7 in semen [39,73]. In reproductive organs, PPV4 was detected in ovaries and uteri [90], suggesting its potential vertical transmission. In this sense, it should be noted that no nPPVs have been reported in colostrum or milk [63]. At the nervous system level, no nPPVs have been found in the central nervous system (CNS), unlike PPV1, which was found in fetal brains in both natural and experimental infections [101], indicating its tissue tropism. The distribution of nPPVs (PPV2 through PPV8) in different organs, excretions, and at the fetal level is compiled in Figure 3. 

The differences in terms of prevalence of these nPPVs between samples and between geographical regions must be affected by various factors such as the epidemiological dynamics of the viruses in each region, the type of sample collected and its handling, along with environmental factors and the techniques used for detection, among others [24,90]. Table 2 shows the prevalence of nPPVs found by samples and the geographical distribution (countries). Regarding the possible inter-species transmission of nPPVs, the only existing report is the detection of PPV3 in beef [102]. There are no reports of transmission to other species, including humans. However, attention should be paid to this issue since there are reports of parvoviruses other than nPPVS. PPV1 was detected in rats [103], and another parvovirus (copivarvovirus) close to PPV4 was found in ticks that affected roe deer [104]. The potential transmission of nPPVs to other species is raised since some of these viruses have been detected in pork intended for human consumption, as is the case of PPV5 and PPV6 detected in pork chops [92] (Figure 3).

## 6. Association of Porcine Parvoviruses with Disease

Members of the *Parvoviridae* family can cause different syndromes, such as enteric, respiratory, reproductive diseases, vascular diseases, and hepatitis, among others. However, of all the PPVs, only PPV1 has been directly associated with a clinical syndrome, SMEDI [4], through techniques such as hemagglutination inhibition assays (HAI) [105,106], serum neutralization (SN), enzyme-linked immunosorbent assay (ELISA) [54,107], real-time PCR [108,109,110], loop-mediated isothermal amplification [111], recombinase-aided amplification (RAA) assay [112], and viral isolation [113,114,115]. Currently, PPV2 is considered potentially pathogenic and a probable participant in PRDC [17,18]. The other nPPVs have unknown pathogenesis since it has not been possible to establish any degree of association between the detection of viral DNA and the clinical signs in the pigs. Additionally, the nPPVs in healthy pigs could indicate that they are simple commensals. These questions remain to be resolved, and particular progress must be made to isolate them and characterize the pathogenesis (if they have it) through both in vitro and in vivo challenges [53].

### 6.1. Porcine Reproductive Failure (PRF)

PPV1 is one of the most relevant reproductive viruses in pig farms. Infection of pregnant sows causes signs such as return to estrus, abortion, mummies, and stillbirths [4], making it a primary agent of SMEDI [69,116]. The consequences of a PPV1 infection depend on the stage of gestation. If infection occurs during the first 70 days, the manifestations are a return to estrus, fetal death, and a decrease in litter size [69]. However, if it occurs after day 70 of gestation, immunocompetent fetuses can survive and be born with anti-PPV1 Abs [116]; strikingly, no clinical signs have been reported in non-pregnant sows or breeding males [4]. Additionally, there are differences in reproductive pathogenesis depending on the infecting PPV1 strain; hypervirulent strains may cause fetal death in the final third of gestation [101]. The association between nPPVs and PRF has been established indirectly since the viral DNA was detected in abortions, being striking, as described before, that nPPVs (except PPV8) were found in the hearts of aborted fetuses [78]. Within these nPPVs, the most prevalent in these samples were PPV4, PPV6, and PPV7, suggesting they could impact the PRF. PPV4 was also detected in the reproductive tissues of sows with PRF and semen [73,88,90]. Interestingly, PPV6 was highly prevalent as a mono-infection in abortions without other PRF-associated pathogens [PCV2, pseudorabies virus (PRV), PRRSV, classical swine fever virus (CSFV), swine influenza virus (SIV), and Brucella)] [12]. In the case of PPV7, this virus was also found in aborted fetuses and farms that reported alterations in reproductive parameters [100,117].

### 6.2. Porcine Respiratory Disease Complex (PRDC)

All nPPVs have been detected in the lungs of pigs with respiratory disease [13,24]. Among these, PPV2 is a potential candidate to be associated with PRDC [17,18]. This attribution is based on the results obtained in several studies determining causality. A serological approach demonstrated high titers of anti-PPV2 Abs in pigs (40–50 days of age) with respiratory disease that were PRRSV-negative and low titers of anti-PCV2 and anti-SIV Abs [118]. Since viral detection, high loads of PPV2 were found in the lungs of pigs with a history of respiratory disease in the fattener and finishing groups [77]. From the pathology perspective, PPV2-positive lungs were associated with lesions such as interstitial pneumonia, broncho-interstitial pneumonia, and pulmonary edema [24]. Finally, since the determination of causality, PPV2-DNA and PPV2-mRNA were detected by in situ PCR and by ISH-RNAscope, respectively, in lung tissue (with different degrees of lesions) precisely in alveolar macrophages and in lymphocytes [17,18] (own unpublished data). Regarding PPV3, this virus was detected at high loads (2.3 × 10^10^ copies/g) in the lungs of pigs with respiratory disease [83]. Likewise, PPV3 and PPV4-DNA were found in lungs with edema [24]. PPV6 is one of the nPPVs found most frequently in the lungs (12.5%) of pigs with PRDC [97]. Finally, PPV5, PPV7, and PPV8 were detected in the lungs of pigs with respiratory symptoms, but their possible association with PRDC has not been established [15,97,100]. An essential contribution to understanding the participation of nPPVs in PRDC was a study that compared the sera from piglets (1–2 months of age) with respiratory symptoms to sera from clinically healthy control pigs and detected PPV2, PPV3, and PPV6 in the sick pigs [97], ultimately reinforcing the proposal that these viruses may be actors of the PRDC.

**Table 2 viruses-15-02398-t002:** Detection and prevalence of the novel porcine parvovirus (PPV2 through PPV7) according to the sample, organs, secretions, and the countries where they have been reported.

	Detection or Prevalence (%) in Different Types of Samples																
	S	L	F	NS	OF	OP	H	T	Sp	LN	K	Li	ABF	Sm	PFET	Country	Reference
PPV2	5.4	21–51	6													Hungary	[17,73]
						25										Romania	[25]
							55	78								Germany	[74]
	19–54		19.4		48.7											Poland	[80,95,119]
	6	30		2				28	43	29–52	100	25	2.8			Europe	[24]
	11–12	2.5–86.6	7.5–12.7							2.5						Republic of Korea	[60,72,98]
	4–8.7	73		0.33	22.5–39.5	22.5										China	[22,58,71,97]
								58–100								Japan	[57]
	22	32				56										Vietnam	[120]
								83								Thailand	[121]
	35–55	20.7–42.7	7.6												7	USA	[11,13,18,122]
															90	Mexico	[78]
						21.8										South Africa	[75]
PPV3	35–48	53.8		10.8	10	22–45		19.5	46–72	71–79		53				China	[10,22,58,82,97]
								73								Thailand	[121]
	16.8	18.4														Vietnam	[120]
	3.8–4.2	7.5–17.5	8.4–10													Republic of Korea	[60,98]
								20								Germany	[74]
	14.4	2.1			5.6	9										Hungary	[73]
		17.5														Romania	[25]
	7.7–15.4		5.6		26.7											Poland	[80,96,119]
	6	6		6					3	3–17	100					Europe	[24]
						21.8										South Africa	[75]
	17.5															DR Congo	[123]
	9.4	9.1														USA	[13]
		65							68.5	57.5	57.5	54				Brazil	[85]
															59	Mexico	[78]
PPV4	5.41	3.85				4.3–51.3										China	[22,58,97]
	3.8–9.5	8.5	2.5–6													Republic of Korea	[60,98]
								44								Thailand	[121]
		7.6														Vietnam	[120]
	2.7	2.3	7										50		50	Hungary	[73]
						17.5										Romania	[25]
								7								Germany	[74]
	2.4–10		13.9		28.7											Poland	[80,95,119]
	3	8		2					7	4–7		15	0.7			Europe	[24]
	5.9	4.1–7.5	4													USA	[11,13]
															25.9	Mexico	[78]
	84	26.6	18.2											38.5		Brazil	[90,124]
						43.6										South Africa	[75]
PPV5	3.4	3–6.6	5.4													USA	[11,13]
	8.2	34.6				9.19										China	[22,97]
	3.8–9.5	21.6	5.6													Republic of Korea	[60,98]
	4–19		21		41.3											Poland	[80,95,119]
															32.4	Mexico	[78]
PPV6	3.8–75	34.6		12.5		3.9							50			China	[12,22,97]
	12.5–19.4	2.5–27.7	2.5–12.4							2.5						Republic of Korea	[60,98]
	6–25.8		17.1		38											Poland	[80,95,119]
	9.4															Rusia	[125]
	13.2															USA	[93]
															74.7	Mexico	[78]
PPV7	2.1	13.8	17.5	17.2												USA	[14]
	24–65	28.3	19.8		17.4	15.4							55			China	[63,96,117,126]
	1.9–35	7.5–74	3.6–22.5										24	1.5		Republic of Korea	[39,98,100]
	15–19.6		39													Poland	[23,119]
												8.6				Brazil	[99]
	21.4		6													Colombia	[62]

Abbreviations of sample tissues, excretions or fluids: L (Lung), K (Kidney), Sp (Spleen), Li (Liver), LN (Lymph Node), S (Serum), F (Feces), H (Heart), T (Tonsils), OP (Organ Pools), NS (Nasal Swabs), OF (Oral Fluid), ABF (Aborted Fetuses), Sm (Semen), PFET (Paraffin and Formalin Embedded Tissues).

### 6.3. Association of Porcine Parvovirus with Other Pathologies

In PPV1, apart from the known effect on PRF, mainly in SMEDI, there are some reports where it was associated with non-suppurative myocarditis [127], enteritis in suckling piglets and young pigs [118,119], and vesicular dermatitis in adult pigs [128]. Hypervirulent PPV1 strains were detected at the CNS level associated with meningoencephalitis lesions, suggesting possible viral replication in the brain [101]. Regarding nPPVs, as already mentioned, viral DNA in sick animal tissues does not indicate causality of infection. Approaches using statistical software led to propose associations between PPV2 and PPV4 with lung pathologies, PPV2 and PPV3 with peritonitis, PPV2 to PPV4 with stomach, colon, and caecum ulcers, PPV4 with renal pathology, and PPV2 with dermatitis and nephropathy syndrome (PDNS) [24]. Additionally, other clinical studies detected nPPVs associated with a variety of pathologies: PPV2 and PPV7 in pericardium pathology [22], PPV2 to PPV4 and PPV7 in neurological disease [22,83,88], PPV2, PPV3, and PPV7 in enteric disease [22,83], PPV3 in lymphadenopathy, PPV3 and PPV4 in lung consolidation, PPV5 in PDNS, PPV6 in liver pathology, and PPV2, PPV3, PPV6, and PPV7 in pigs with renal pathology [22]. With the above, it is clear that studies should be proposed that seek to explain these associations. However, the most significant limitation will continue to be the impossibility of conducting controlled studies due to the need for viral isolates.

### 6.4. Viral Coinfections between Porcine Parvovirus and Other Viruses

In pig farms, reports of coinfections between endemic circulating viruses and emerging ones are becoming more common [129]. As it is a recent topic, the impact of these concurrences on the pathogenesis and clinical symptoms is still unclear. In vitro and in vivo tests have shown that the simultaneous presence of two or more viruses in a cell or an individual can cause effects such as (i) viral interference against one of the infecting viruses, (ii) an increase in the replication or virulence of one of them or both, and (iii) recombination events if they are genetically close viruses, among others [129]. Coinfections involving PPVs are frequent and mainly concur with PCV2 and PRRSV, making PPVs probably participate in PCVAD or PRDC-PRF, respectively [13,97]. Reports show that PPV1 can increase the severity of PCV2 [130] and favor the development of PMWS [131,132]. PPV1/PCV2 coinfection was associated with more severe cardiac lesions in stillbirths than those infected with PCV2 or PPV1 alone [127,133]. Regarding nPPVs, several studies evaluate coinfections where these viruses are present in pigs with PCVAD, PMWS, or in PRF [13,78,97], as well as the association of nPPVs with subclinical infection of PCV2, PCV3, and PRRSV [60,119] (own unpublished data). Evidence shows that PPV2 is the nPPV most detected in coinfection (probably because it is the most diagnosed) and was associated, as already discussed, with PRDC. The most frequent coinfections found have been PPV2/PCV2 [17,18], PPV2/PRRSV [18,60], PPV2/SIV [18], and PPV2/M. hyopneumoniae. Something striking is that the highest detection of PPV2 has been found in PCV2-positive pigs with PCVAD [13,97]. However, PCV2/PPV2 coinfection in pigs vaccinated against PCV2 did not show any consequence on virulence regarding clinical signs or more severe histopathological lesions [134]. In the case of PPV3, this virus has been detected in pigs with PRDC in coinfections such as PPV3/PCV2, PPV3/PRRSV, PPV3/SIV and PPV3/M. hyopneumoniae [83,97], particularly with PPV3/PCV2 concurrence, was associated with PMWS and PCVAD clinical manifestations [58,60,82]. PPV4 was detected in PPV4/PCV2 coinfection in pigs with PCVAD [33,58,124], also in sows with PRF in PPV4/PCV2, PPV4/TTV1, and PPV4/TTV2 coinfections [88]. For PPV5, it was found in PPV5/PCV2 and PPV5/PRRSV coinfections associated with PRDC [60] and PPV5/PCV2 associated with PRF [78]. PPV6 was detected in pigs with PRDC [97] or PCVAD [60] and PRRSV/PPV6 coinfection in sera [93]. Regarding detecting nPPVs in concurrence with PCV2-subclinical disease (PCV2-SD), this was found in growing pigs, the most frequent being PCV2/PPV3, PPV5, and PPV6 [119]. While in PCV3-SD sows, the highest coinfection was with PPV6 (own unpublished data). Regarding PRRSV, subclinical infection was associated with PPV5 (own unpublished data). PPV7-positive cases were higher in farms [97] and PCV2-positive pigs [135]. In the cases of PRF, PPV7 was primarily detected in PCV3-positive sows [117]. In addition to this, high viral loads for PCV2 and PCV3 were found in PPV7-positive pigs [63,117,119]. The above may suggest that PPV7 could stimulate the replication of PCVs and be a cofactor for the presentation and severity of PCVAD and PCV3 [117,119]. Likewise, higher viral titers to PRRSV have been determined in PPV7-positive pigs [60,63]; this suggests that PPV7 would also stimulate PRRSV replication [63]. Regarding PPV8, there is only one report about coinfection in lung tissue concurrently with PRRSV [15]. Summarizing what was stated in this section, many studies corroborate the presence of nPPVs in coinfection in healthy and sick pigs (mainly PRDC and PRF) with other primary viral agents. It is entirely unknown if nPPVs increase the severity of clinical signs, if they favor the replication of other viruses, or if they have no effect. The main coinfections between nPPVs and other swine viruses are described in Table 3.

**Table 3 viruses-15-02398-t003:** Coinfections detected between the novel porcine parvovirus and other porcine viruses.

Novel Porcine Parvovirus	Coinfections	Reference
PPV2	PCV2	[13,24,60,73,78,97,119,122]
	PRRSV	[18,60,97]
	PPV1 to PPV7	[25,75,78,98]
	PRV	[97]
	SIV	[18]
	PTTV1	[97]
	PBo-like V, PBoV	[75]
	*M. hyopneumoniae*	[18]
PPV3	PCV2	[13,24,73,78,82,97,119]
	PRRSV	[60,97]
	PPV1 to PPV7	[75,78,98]
	PRV	[97]
	PTTV1	[97]
	PBo-like V, PBoV	[75]
	SIV	[83]
	*M. hyopneumoniae*	[83]
PPV4	PCV2	[13,78,88,97,119]
	PRRSV	[60]
	PPV1 to PPV7	[75,78,98]
	PBo-like V, PBoV	[75]
	PTTV1, PTTV2	[88,97]
PPV5	PCV2	[13,78,97,119]
	PRRSV	[60,97]
	PRV	[97]
	PTTV1	[97]
	PPV1 to PPV7	[78,98]
PPV6	PCV2	[13,78,97,119]
	PRRSV	[60,93,97]
	PPRV	[97]
	PPV1 to PPV5 and PPV7	[78,98]
PPV7	PCV2	[63,96,119]
	PRRSV	[60,63]
	PCV3	[63,117]
	PEDV	[62]
	PPV2 to PPV6	[98]
PPV8	PRRSV	[15]

Abbreviations: PRRSV (porcine respiratory and reproductive syndrome virus), PCV (porcine circovirus), PPV (porcine parvovirus), SIV (swine influenza virus), PBo-likeV (porcine boca-like virus), PBoV (porcine bocavirus), PTTV (Porcine Torqueteno virus), PEDV (porcine epidemic diarrhea virus), PRV (porcine pseudorabies virus), and M (Mycoplasma).

## 7. Conclusions

Until the beginning of this century, PPV1 was known as the only infectious PPV in the swine population. Seven others have been discovered whose participation as triggering disease agents is the subject of research. These nPPVs are known to have a wide distribution worldwide, have a high prevalence, and be present in both domestic and wild pigs. Although the origin of these viruses has not been established, phyloevolution studies indicate that nPPVs have been circulating long before their first reports and have been evolving with high mutation rates. Particularly in wild pigs, recombination events have been reported between these new species, making these wild swine populations the subject of study on this topic. 

Regarding tropism and transmission, nPPVs have been detected in various organs and tissues; from this, it is inferred that its tropism is also very broad. Likewise, effective horizontal and vertical transmission has been suggested, the first proposed by the viral presence in almost all secretions and the second by detecting nPPVs in aborted fetuses. Regardless of the above, whether they are pathogenic is debatable. The fact that they have been detected in sick and healthy pigs has led to the proposal of two hypotheses: one that proposes that they are part of the virome of pigs and another that they are involved as triggering agents or facilitators of disease. The answer to the above will be obtained when its isolation is achieved, and in vitro and in vivo pathogenesis tests can be carried out. Additionally, these nPPVs probably do not escape the biological and physical characteristics of the rest of the PVs, which means that they are very resistant in environments where they can last for long periods, which would contribute to their lasting longer, which can colonize other hosts.

At the farm level in swine production systems, nPPVs have been detected with a higher prevalence in the fattener and finishing pig groups (above 12–14 weeks of age) and a lower prevalence in piglets. The above suggests that pigs are infected within the mentioned groups’ farms and are protected in the first weeks of life. Pigs have indeed received passive antibodies, and the curve of these gradually declines until the pigs become infected, or if they are subclinical, a reactivation of viral replication occurs. There are more questions than answers about this, and they need resolution. The questions are: If there are antibodies in piglets, the sows have the viruses, and where do they come from? At what point is the progeny infected, in the uterus, with colostrum? The few reported approaches demonstrate (no virus detected in colostrum) that this last route is ineffective. What are the antibody dynamics like in piglets? All this is yet to be resolved.

From the pathogenesis, the compilation of the published results shows that an association of nPPVs with some syndromes can be proposed, mainly with respiratory and reproductive disease, even though, as is the case of PPV2, it can be suggested that they generate disease. It is necessary to continue searching for associations between the presence of nPPVs and the manifestation of clinical signs and pathological lesions. Finally, it is also vital to evaluate the impact of nPPVs on coinfections by determining whether they contribute to the pathogenicity of other primary viruses.

## Figures and Tables

**Figure 1 viruses-15-02398-f001:**
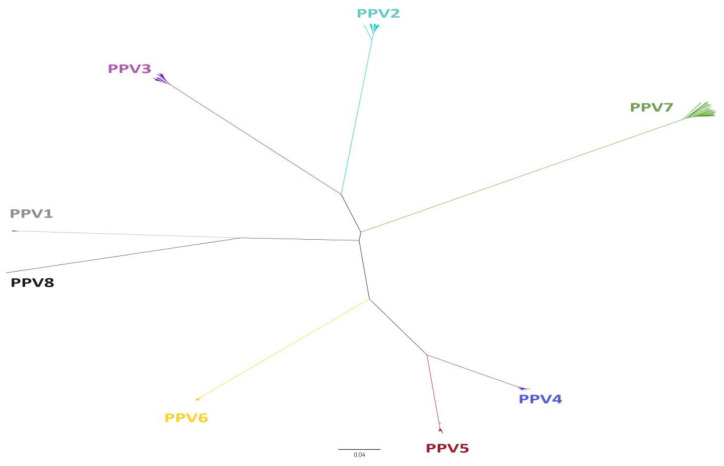
Phylogenetic distribution of porcine parvoviruses (PPVs). It was inferred based on complete nucleotide sequences of the non-structural gene (NS) selected from representative strains of PPV1 through PPV7 obtained from the GenBank. The evolutionary tree was constructed with the neighbor–joining method by the p-distance model. The analysis involved 201 nucleotide sequences. Evolutionary analyses were conducted in MEGA7 [28].

**Figure 2 viruses-15-02398-f002:**
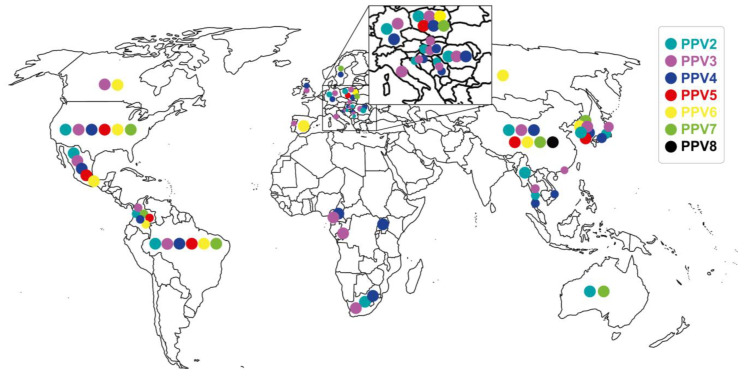
Global distribution of novel porcine parvoviruses (nPPVs), PPV2 through PPV8. Graph prepared on the geographical location of the nPPVs in literature reports and the origin of the sequences deposited and accessed from GenBank.

**Figure 3 viruses-15-02398-f003:**
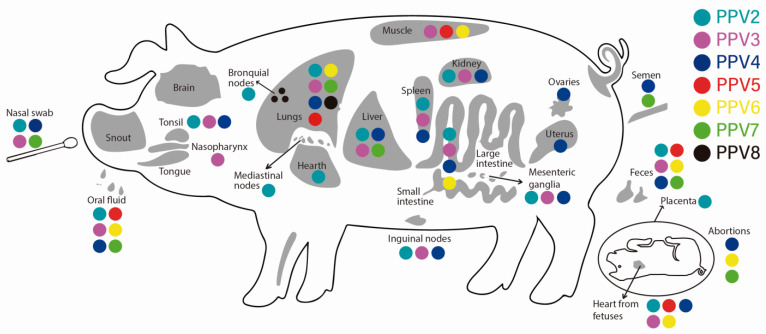
Detection of the novel porcine parvovirus (PPV2 through PPV8) in organs, excretions, and abortions of pigs, according to the reports found in the literature.

## Data Availability

All required data are available as texts and figures in the main text of the article. The sequence datasets are publicly available at NCBI GenBank.

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
