# Peer review of "The Novel Porcine Parvoviruses: Current State of Knowledge and Their Possible Implications in Clinical Syndromes in Pigs"

_viruses, 2023, doi:10.3390/v15122398_

Round 1

Reviewer 1 Report

Comments and Suggestions for Authors

This paper provides an in-depth review of PPVs. It will undoubtedly offer valuable information to related researchers. However, it would be even more excellent if the following content is further supplemented. 

The authors suggest that one of the reasons why PPV can appear in various forms is due to virus recombination within wild boars. However, if one points to wild boars as a cause of diversification into various strains of a virus, a deeper review on this would offer more profound insights into the diversification of PPVs. Such content should, of course, also be reflected in the abstract.

Reviewer 2 Report

Comments and Suggestions for Authors

Comments and Suggestions for Authors

It is well known that porcine parvovirus (PPV) cause various diseases and coses economic losses to global swine industry. To date, eight different porcine parvoviruses, PPV1 and seven called novel PPVs (nPPV2- nPPV8) were detected in the swine population. The pathogenesis of nPPVs is still undefined, and its infectious potential is unknown. The study design, data collection section and the approach to deduce the results/conclusion are well prepared. I don't have comments regarding style and language.

 The paper is very readable, except Table 2. (Detection and prevalence of the novel porcine parvovirus (PPV2 through PPV7) according to the samples, organs, secretions, and the countries where they have been reported). Vertical table will be easier to understand.

Reviewer 3 Report

Comments and Suggestions for Authors

Comments/suggestions according to row number:

- 50: can you add a reference re. to Invertebrates ?

- 51 to 54: it would be more clear to add the species involved in the indicated pathologies (e.g: dog; cat; etc)

- 55 : add at least one reference for this statement (can you anticipate here ref. 24, 23 ?)

- 68: perhaps it would be more clear " no nPPVs have been isolated yet" or similar...

- 268: "spread" or "was demonstrated/detected" (like, for example,  in  row 273) ? "sperad may mean that it was demonstrated that there was, indeed, a spread of the virus thru (?) semen, material, biological samples, air ..... ???

- 370: table 2 is very interesting, but difficult to read in this way: any possibility to reorganize in a "landscape" view ? and, if possibible, on a single page ?

- 379: "associated thru....." (indicate shortly trhu which laboratory tests and, first of all, isolation, replication.... this is the main concept/difference you are mentioning and insisting on )

Reviewer 4 Report

Comments and Suggestions for Authors

Dear editor, 

In my opinion the review submitted by Diana S. Vargas-Bermudez et al., is a very well written and organized manuscript that can be used by researchers in the field of porcine diseases. The review covers a lot of references and information about porcine parvoviruses starting with classification and molecular organization, Evolution, Epidemiology, Tropism and  association with diseases. 

I recommend this manuscript for publication in the present form. 
